# Hsp70 Negatively Regulates Autophagy via Governing AMPK Activation, and Dual Hsp70-Autophagy Inhibition Induces Synergetic Cell Death in NSCLC Cells

**DOI:** 10.3390/ijms25169090

**Published:** 2024-08-22

**Authors:** Bashar Alhasan, Yana A. Gladova, Dmitry V. Sverchinsky, Nikolai D. Aksenov, Boris A. Margulis, Irina V. Guzhova

**Affiliations:** Lab of Cell Protection Mechanisms, Institute of Cytology, Russian Academy of Sciences, 194064 St. Petersburg, Russia; st085770@student.spbu.ru (Y.A.G.); sverchinsky@incras.ru (D.V.S.); and@incras.ru (N.D.A.); margulis@incras.ru (B.A.M.); irina.guzhova@incras.ru (I.V.G.)

**Keywords:** cancer, proteostasis, autophagy, Hsp70, drug resistance, combinatorial cancer therapy, AMPK, mTOR

## Abstract

Proteostasis mechanisms, such as proteotoxic-stress response and autophagy, are increasingly recognized for their roles in influencing various cancer hallmarks such as tumorigenesis, drug resistance, and recurrence. However, the precise mechanisms underlying their coordination remain not fully elucidated. The aim of this study is to investigate the molecular interplay between Hsp70 and autophagy in lung adenocarcinoma cells and elucidate its impact on the outcomes of anticancer therapies *in vitro*. For this purpose, we utilized the human lung adenocarcinoma A549 cell line and genetically modified it by knockdown of Hsp70 or HSF1, and the H1299 cell line with knockdown or overexpression of Hsp70. In addition, several treatments were employed, including treatment with Hsp70 inhibitors (VER-155008 and JG-98), HSF1 activator ML-346, or autophagy modulators (SAR405 and Rapamycin). Using immunoblotting, we found that Hsp70 negatively regulates autophagy by directly influencing AMPK activation, uncovering a novel regulatory mechanism of autophagy by Hsp70. Genetic or chemical Hsp70 overexpression was associated with the suppression of AMPK and autophagy. Conversely, the inhibition of Hsp70, genetically or chemically, resulted in the upregulation of AMPK-mediated autophagy. We further investigated whether Hsp70 suppression-mediated autophagy exhibits pro-survival- or pro-death-inducing effects via MTT test, colony formation, CellTiter-Glo 3D-Spheroid viability assay, and Annexin/PI apoptosis assay. Our results show that combined inhibition of Hsp70 and autophagy, along with cisplatin treatment, synergistically reduces tumor cell metabolic activity, growth, and viability in 2D and 3D tumor cell models. These cytotoxic effects were exerted by substantially potentiating apoptosis, while activating autophagy via rapamycin slightly rescued tumor cells from apoptosis. Therefore, our findings demonstrate that the combined inhibition of Hsp70 and autophagy represents a novel and promising therapeutic approach that may disrupt the capacity of refractory tumor cells to withstand conventional therapies in NSCLC.

## 1. Introduction

Cancer cells employ multiple protective mechanisms to confront the frequent changes in microenvironmental conditions and survive the toxic effects of various stress factors, including hypoxia, starvation, therapeutic interventions, and immune response [1,2,3]. These stress-inducing conditions disturb cellular homeostasis and elicit proteotoxic stress, leading to the activation of protein homeostasis (or proteostasis) machineries, including proteotoxic-stress response (PSR), autophagy, ubiquitin-proteasome system (UPS), and unfolded protein response (UPR), which collectively impact several tumor hallmarks such as proliferation, apoptosis, drug resistance, and metastasis [4,5,6].

The members of the Hsp70 (HSPA) family constitute the most abundant heat-shock proteins, whose synthesis is elevated as a part of the proteotoxic-stress response (or heat-stress response (HSR)) following the activation of heat-shock factor 1 (HSF1) [7,8]. These proteins play a crucial role in recognizing and refolding newly synthesized or damaged polypeptides or directing incorrigible damaged structures to ubiquitination and subsequent proteasomal degradation [9].

Macroautophagy (hereafter referred to as autophagy) is an evolutionarily conserved proteostasis mechanism utilized by eukaryotic cells at basal levels under normal settings or at elevated levels during stress to eliminate misfolded proteins, large protein aggregates, and damaged organelles inaccessible to smaller proteolytic systems [10,11]. Autophagy is primarily regulated by the interplay between two protein kinases: the mechanistic target of rapamycin (mTOR), which inhibits autophagy by suppressing Unc-like kinase 1 (ULK1) activity [12], and the adenosine-monophosphate dependent protein kinase (AMPK), which, in contrast, induces autophagy by activating ULK1 and inhibiting mTOR activity [13].

In cancer cells, both autophagy and Hsp70 are often engaged and exhibit powerful cytoprotective roles that promote survival under various stresses, including anti-cancer therapies, by compromising intrinsic and extrinsic apoptotic pathways, interfering with proliferation, dormancy, immune response, and metastasis [14,15,16]. Consequently, combinatorial therapeutic approaches employing inhibitors of Hsp70 or autophagy have attracted significant research interest for their potential to sensitize cancer cells to therapeutic agents [17,18,19]. For example, multiple lines of evidence, including our own, have demonstrated that Hsp70 inhibitors in combination regimens markedly enhance the sensitivity of various tumor cell types to conventional therapies such as doxorubicin, etoposide, and bortezomib, both *in vitro* and *in vivo* [20,21,22,23]. Likewise, early-stage (ULK1 or VPS34 inhibitors) or late-stage autophagy inhibitors (chloroquine (CQ) and hydroxychloroquine) have exhibited remarkable antitumor effects in several cancer models, some of which underwent or are currently undergoing clinical trials [24,25,26,27]. Additionally, we have recently reported the promising efficacy of simultaneously targeting autophagy and HSF1 in primary colorectal cancer cells derived from patients [28].

Nevertheless, despite extensive research, the crosstalk between Hsp70 and autophagy in cancer cells, particularly in the context of therapeutic resistance, remains insufficiently understood. Hence, in this work, we aimed to investigate the molecular interplay between Hsp70 and autophagy in lung cancer, the most fatal cancer type worldwide, and to develop an effective combinatorial therapeutic approach based on their reciprocal interaction in non-small cell lung cancer (NSCLC) cells.

## 2. Results

### 2.1. Autophagy Is Differentially Regulated upon Heat Shock Exposure or Hsp70 Chemical Hyperactivation in NSCLC Cells

To investigate how proteotoxic-stress response and its major molecular chaperone Hsp70 may interfere and regulate the activation of autophagy in NSCLC tumor cells, we first explored the effects of HSF1-mediated overactivation of Hsp70 on autophagy levels via two different approaches: exposing tumor cells to heat shock at 43 °C for 1 h or employing the chemical inducer of Hsp70 synthesis, ML346 [29]. Following the heat shock exposure, tumor cells were incubated for either 5 h or 18 h to allow recovery before being lysed to assess changes in autophagy levels. These recovery periods were selected based on previous reports and our own experiments on the dynamics of Hsp70 accumulation in tumor cells following heat shock exposure, with a notable accumulation observed starting from 3 h of recovery after heat shock treatment and a maximal accumulation occurring between 12 and 24 h [30,31,32,33]. The autophagic flux was evaluated by measuring the expression level of the key autophagic markers, p62 degradation, and LC3 I/II conversion.

As anticipated, the exposure of tumor cells to heat shock induced the expression of Hsp70 in both A549 and H1299 cell lines, which was markedly upregulated (3–4-fold) after 18 h of recovery (Figure 1A–D). Consequently, this increase in Hsp70 expression upon the heat shock was accompanied by enhanced autophagic flux, as evidenced by the increased conversion of LC3-I to LC3-II with reduced levels of p62, indicating its degradation by autophagy and increased autophagic flux (Figure 1A–D). Strikingly, the chemical induction of Hsp70 using ML346 (10 µM) resulted in opposite effects on autophagy since a reduction in LC3 I/II conversion was observed while p62 was accumulated (Figure 1E,F), suggesting a differential regulation of autophagy following the overactivation of PSR and Hsp70 by the exposure to heat shock or by the chemical inducer ML346 in NSCLC cells.

### 2.2. HSF1 Knockdown and Genetic or Chemical Inhibition of Hsp70 Activate Autophagy, While Genetic Hsp70 Overexpression Suppresses Autophagy in NSCLC Cells

To delineate the precise impact of the pivotal PSR regulator, HSF1, and the major chaperone, Hsp70, on autophagy activity, we generated A549 cell lines with knockdown of HSF1 or Hsp70. The efficient knockdown of Hsp70 or HSF1 was validated via western blotting, where the protein levels of Hsp70 and HSF1 were significantly reduced, respectively (Figure 2A,B and Appendix A). Importantly, we observed augmented levels of ATG5 and LC3 I/II conversion following the knockdown of both HSF1 and Hsp70, concomitant with diminished levels of p62, indicating that autophagy is activated subsequent to HSF1 and Hsp70 genetic knockdown and that this autophagic induction is dependent to a certain extent on the downregulation of Hsp70 (Figure 2A,B).

To corroborate these findings, we pharmacologically inhibited the activity of Hsp70 by treating A549 and H1299 tumor cells with increased concentrations of VER-155008 for 24 h, a selective and potent inhibitor of Hsp70 function, by competitively binding its ATPase binding domain [34]. Immunoblotting analysis revealed no discernible impact on intracellular Hsp70 levels following VER-155008 treatment (Figure 2C,D,F,G); however, a reduction in Hsp70 function was evident through its diminished capacity to bind the denatured HSP70 substrate, carboxymethylated lactalbumin (CMLA), in a concentration-dependent manner, as assessed by substrate-binding ELISA (Figure 2E). Consequently, we observed a concentration-dependent increase in autophagic flux upon Hsp70 inhibition by VER-155008, as evidenced by the elevated levels of ATG5, conversion of LC3-I to LC3-II, and notable degradation of p62 (Figure 2C,D,F,G). This upregulation of autophagy function upon genetic or chemical suppression of Hsp70 also involves transcriptional upregulation of autophagy-related genes, ULK1, p62, and ATG7, as demonstrated by RT-PCR results (Appendix A). Furthermore, given that Hsp70 inhibitors bind the chaperone on different sites and suppress its function by distinct mechanisms, the autophagic induction following Hsp70 ablation was recapitulated using another Hsp70 inhibitor, JG-98 [35] (Appendix A). The enhanced autophagic flux was further validated by treating tumor cells with VER-155008 in the presence of CQ, which also revealed a significant increase in LC3-II accumulation upon combined treatment compared to CQ treatment alone (Appendix A).

Consistent with these findings, Hsp70 knockdown in H1299 cells resulted in a similar induction of autophagy as observed in A549 shHsp70 cells. Furthermore, we generated an H1299 cell line with a genetic Hsp70 knock-in in these cells. Interestingly, genetic Hsp70 overexpression in these cells exhibited inhibitory effects on autophagy, as evidenced by the increased accumulation of LC3-I and p62 (Figure 2H,I), consistent with the results obtained following the overactivation of Hsp70 by ML-346. Collectively, these data, in conjunction with prior observations, implicate a suppressive role of the HSF1-mediated PSR on autophagy, with Hsp70 serving as a significant mediator of this negative regulatory mechanism.

### 2.3. Rapamycin-Induced Autophagy Downregulates Hsp70 Levels, While Inhibition of Autophagy by SAR405 Does Not Influence the Levels of Hsp70 in NSCLC Cells

The next step was to investigate whether modulating autophagy activity could influence Hsp70 expression levels or function. For this purpose, we initially induced autophagy by treating tumor cells with rapamycin, the well-known autophagy inducer that acts by inhibiting mTOR kinase activity, at a concentration of 100 nM for 2 h. After that, tumor cells were exposed to heat shock for 1 h, followed by an 18 h recovery period, after which the levels of Hsp70 and autophagy-related markers were quantified by immunoblotting. Intriguingly, pretreatment of tumor cells with rapamycin and subsequent autophagy induction led to a slight reduction in the levels of Hsp70 in normal conditions and, furthermore, compromised its upregulation subsequent to exposure to heat shock in both A549 and H1299 cell lines (Figure 3A–D).

Afterwards, we sought to elucidate the effects of autophagy inhibition on Hsp70 levels. Autophagy suppression was initially achieved using SAR405, a potent and selective inhibitor of PIK3C3/VPS34 that impedes the formation of autophagosomes and the subsequent stages of autophagy. Accordingly, A549 tumor cells were treated with SAR405 at increased concentrations (5, 10, and 20 µM) for 24 h, followed by evaluation of autophagy and Hsp70 levels via immunoblotting. Consequently, SAR405 treatment resulted in a potent inhibition of the autophagic flux, as evidenced by a robust accumulation of p62 and LC3-I, indicative of the suppressed conversion of LC3-I to LC3-II (Figure 3E,G–I). Importantly, no significant changes in Hsp70 activation levels were detected upon autophagy suppression by SAR405 (Figure 3E,G–I). Moreover, assessment of Hsp70 functionality via substrate-binding assay indicated that SAR405 treatment did not affect the functional capacity of Hsp70 or its ability to bind the denatured substrate CMLA (Figure 3F). Similar findings were observed following autophagy inhibition by CQ (Appendix A).

### 2.4. Hsp70 Orchestrates the Activation of Autophagy by Upregulating mTOR, Suppressing AMPK and ULK1/Beclin1 Activation

Canonical macroautophagy is tightly regulated by the functional interplay of AMPK and mTOR activities. mTOR activation has been elucidated to inhibit autophagy by restraining AMPK and phosphorylating ULK1 at Ser757, thereby deactivating ULK1 [12]. Conversely, AMPK activation potentiates autophagy by activating TSC2 and inhibiting RAPTOR, which inhibits mTOR activation, and also by phosphorylating ULK1 at Ser555 and Ser317, along with activating Beclin1 [13,36] (Figure 4A). Hence, we asked whether the Hsp70-mediated regulation of autophagy might be exerted through influencing AMPK/mTOR activity.

To assess this hypothesis, we investigated how the modulation of Hsp70 activation impacts the activity of the AMPK/mTOR axis. Starting with chemical overactivation by ML346 and Hsp70 genetic knock-in, we observed a significant increased phosphorylation of mTOR at Ser2448, coupled with a reduction in AMPK phosphorylation (Figure 4B,C,F,G). Furthermore, we observed that the total ULK1 protein levels were reduced, concomitant with increased phosphorylation at Ser757 by mTOR and decreased phosphorylation at Ser555 by AMPK (Figure 4B,C). In addition, reduced levels of Beclin1 were evident upon Hsp70 genetic hyperactivation in H1299 cells (Figure 4F,G).

To corroborate these findings, we evaluated the impact of genetic Hsp70 downregulation or chemical inhibition using VER-155008. Accordingly, knockdown of Hsp70 augmented the phosphorylation of AMPK, abrogated mTOR activation, and increased Beclin1 levels (Figure 4F,G). Furthermore, the total levels of ULK1 and its phosphorylation by AMPK at Ser555 were documented upon the knockdown of Hsp70 or HSF1 in A549 tumor cells (Figure 4D,E). In line with these data, Hsp70 inhibition by VER-155008 diminished mTOR phosphorylation, enhanced AMPK phosphorylation, upregulated total and phosphorylated ULK1 at Ser555, as well as increased Beclin1 levels (Figure 4H,I and Appendix A). Taken together, these findings indicate that Hsp70 regulates autophagy by governing the activation of AMPK/mTOR kinases, which impacts the subsequent recruitment and activation of the major autophagic proteins, ULK1 and Beclin1, in NSCLC cells.

Despite the pronounced activation of Hsp70 during heat shock exposure, the mechanism by which autophagy was triggered remains uncertain. To shed light on this process, we examined the phosphorylation levels of AMPK and mTOR, along with the protein levels of ULK1 and Beclin1, upon heat shock exposure, both with and without rapamycin treatment. Our findings indicated that heat shock exposure and Hsp70 upregulation were associated with decreased AMPK phosphorylation and even hindered its activation by rapamycin (Figure 4J,K). Similar inhibitory effects were documented with total and phosphorylated levels of ULK1, while heat shock-induced Hsp70 was accompanied by a modest increase in mTOR phosphorylation and reduced inhibition by rapamycin (Figure 4J,K). Notably, Beclin1 levels were significantly elevated under heat shock, rapamycin treatment, or their combination (Figure 4J,K), suggesting that heat shock-induced autophagy is AMPK/ULK1-independent but noncanonical Beclin1-dependent autophagy.

### 2.5. The Combined Inhibition of Hsp70 and Autophagy Synergistically Enhances Cisplatin-Induced Apoptotic Cell Death in NSCLC Cells

Autophagy has dual roles in cancer, acting as either a pro-survival or pro-death cellular mechanism, which was shown to be context- and tumor type-dependent. Building upon our elucidation of the autophagy upregulation upon Hsp70 inhibition, we aimed to determine its impact on tumor cell viability. Additionally, we explored how this autophagy role could be leveraged in combinatorial regimens to sensitize NSCLC cells to conventional therapies. To this end, treating A549 tumor cells with cisplatin, a standard first-line chemotherapy agent for lung cancer, alone or in combination with VER-155008, promoted the transcriptional upregulation of both Hsp70 and autophagy-related genes, ULK1, ATG7, and p62 (Appendix A), suggesting the upregulation of autophagy and Hsp70 to confront cisplatin cytotoxicity [30,37]. Afterwards, a series of cell viability assays using the MTT test were conducted, employing various combinations of Hsp70 inhibitors VER-155008 (10 µM), JG-98 (1 µM), autophagy modulators SAR405 (10 µM), CQ (40 µM), and rapamycin (0.1 µM) with cisplatin (10 µM) for 48 h. The concentrations of the compounds were selected based on prior MTT assays, whereby the selected concentration of each concentration caused no more than 20–25% cell death.

Consequently, our results revealed that the concurrent inhibition of Hsp70 and autophagy using VER-155008 and SAR405, respectively, significantly enhanced the cell death of A549 and H1299 tumor cells. Importantly, dual targeting of Hsp70 and autophagy with cisplatin resulted in a substantial induction of tumor cell death (Figure 5A–D), with the observed effects resembling a synergetic interaction. Similar observations were obtained using the Hsp70 inhibitor JG-98 and the well-known autophagy inhibitor CQ (Figure 5A–D). Interestingly, substituting SAR405 or CQ with rapamycin yielded lower efficacy, with a slight or non-significant enhancement in tumor cell death (Figure 5A–D), as for example, the combination of cisplatin,VER-155008, and rapamycin on A549 tumor cells (Figure 5A).

To assess whether the observed effect of combining cisplatin with Hsp70 and autophagy inhibitors was synergistic, we calculated the drugs’ combination index (CI) using the formula CI = ABC/(A + B + C). Here, ABC denotes the fraction of killed cells after treatment with the three drugs combination, normalized to the control group (untreated cells), while A, B, and C represent the fraction of killed cells after treatment with a single drug treatment, normalized to the control group, respectively. A CI value of <1 indicates synergism among the three drugs, with CI values ranging from 0.7 to 0.9, indicative of moderate synergism, while a CI value of >1 indicates antagonism. Notably, the CI value for the concurrent treatment of cisplatin, VER-155008, and SAR405 on A549 tumor cells was 0.7, suggesting a synergistic effect from this combination. In the case of using rapamycin instead of SAR405, the value of CI was 1.1, suggesting a slight antagonistic effect from this combination.

Given that the MTT assay primarily assesses the metabolic activity of tumor cells rather than cell viability, we performed the Annexin V/PI staining apoptosis assay to validate our findings and determine the mechanism by which dual Hsp70 and autophagy suppression may exert its synergetic induction of cell death. Our results indicated that the combination of cisplatin with either VER-155008 or SAR405 alone augmented cisplatin-induced cell death, while the simultaneous treatment of all three compounds induced remarkable cell death and substantially potentiated apoptosis, with the observed effect again indicative of a synergistic one (Figure 5E–H). Strikingly, the use of rapamycin instead of SAR405 rescued tumor cells from apoptosis, compared to the cisplatin and VER-155008 combination in NSCLC cells (Figure 5E–H).

Collectively, these observations suggest that autophagy plays a pro-survival role upon Hsp70 inhibition, acting predominantly as a compensatory mechanism to alleviate proteotoxic stress. Accordingly, dual targeting of Hsp70 and autophagy, in combination with cisplatin, appears to exhibit synergism and be more efficacious in inducing tumor cell death than utilizing autophagy activators. These findings were also corroborated by colony formation assays, demonstrating that the combined inhibition of Hsp70 and autophagy significantly enhanced the efficacy of cisplatin in reducing the ability of tumor cells to proliferate and form colonies compared to utilizing rapamycin (Figure 6A and Appendix A).

### 2.6. Dual Targeting of Hsp70 and Autophagy Provokes Substantial Cell Death in Three-Dimensional (3D) Spheroids of NSCLC Cells

While two-dimensional (2D) cell cultures are suitable for high-throughput drug toxicity screening, it is essential to employ pre-clinical models that more accurately mimic *in vivo* tumor biology for a precise assessment of therapeutic toxicity and efficacy. Therefore, we utilized an *in vitro* three-dimensional (3D) spheroid model of NSCLC cells, as 3D spheroids better replicate certain characteristics of solid tumors, including gene expression profiles, secretion of soluble mediators, and mechanisms of chemoresistance [38]. After cultivating for 3 days, 3D spheroids were treated with different combinations of Hsp70 inhibitors and autophagy modulators with cisplatin. Then, the morphology of 3D spheroids was captured every two days, and their viability was evaluated on day six post-treatment using the TiterGlo 3D cell viability assay.

Accordingly, we found again that cisplatin treatment, concurrently with Hsp70 and autophagy inhibitors, resulted in the most significant destruction of spheroidal morphology (Figure 6B,D), accompanied by the highest level of induced cell death, as indicated by the TiterGlo assay (Figure 6C,E). Notably, substituting SAR405 with rapamycin did not completely disrupt spheroid morphology and did not diminish spheroid cell viability (Figure 6B–E). These findings further highlight the significance of Hsp70 and autophagy in enabling lung cancer cells to withstand chemotherapy-induced cytotoxicity, and provide a rationale for the combined inhibition of Hsp70 and autophagy to synergistically impede cancer cell tolerance to chemotherapeutic agents and enhance their efficacy in eradicating tumor cells.

## 3. Discussion

The cellular proteostasis systems, including heat-shock proteins and autophagy, are implicated in various aspects of tumor biology like tumorigenesis, drug resistance, metastasis, and recurrence [6,15,39]. Therefore, a better understanding of their involvement in distinct tumor manifestations, along with their reciprocal regulation and interplay, is crucial to improving the clinical outcomes of cancer therapies. In this study, we explored the interaction of two major components of proteostasis systems, Hsp70 and autophagy, and based on their crosstalk, we proposed a combinatorial therapeutic approach that may effectively disrupt the capacity of tumor cells to withstand conventional therapies.

Our results demonstrate that the major PSR regulator, HSF1, negatively regulates autophagy activation in NSCLC cells, as evidenced by the increase in autophagic flux following HSF1 knockdown or the opposite effects upon its activation by ML346. This negative regulation could be linked to HSF1 crosstalk with mTOR and AMPK functions, as HSF1 enhances mTOR activation by suppressing JNK-mediated inhibition of mTORC1, while mTORC1 phosphorylates HSF1 at Ser326 to promote its function [40,41,42]. Conversely, a repressive reciprocal regulation was documented between HSF1 and AMPK, whereby HSF1 physically impairs AMPK phosphorylation while AMPK phosphorylates HSF1 at Ser121, resulting in HSF1 inhibition under metabolic stress [43,44].

In addition, we demonstrated that PSR-mediated negative regulation of autophagy involves the activity of Hsp70. Specifically, the downregulation of Hsp70, whether resulting from HSF1 knockdown, direct Hsp70 knockdown, or chemical inhibition of Hsp70 function by VER-155008 or JG-98, was associated with increased autophagy activation. Conversely, chemical or genetic hyperactivation of Hsp70 decreased the levels of autophagy-related proteins and the autophagic flux.

In mechanistic exploration, we found that Hsp70 overexpression was directly associated with upregulated mTOR activation, reduced AMPK phosphorylation, and decreased ULK1 levels with enhanced phosphorylation at Ser757 by mTOR, which collectively represent autophagy-inhibiting events [12]. Conversely, HSF1 or Hsp70 knockdown resulted in enhanced phosphorylation of AMPK and increased levels of ULK1 and its phosphorylation by AMPK at Ser555, which triggers autophagy induction [13]. Additionally, we demonstrated that Hsp70 inhibition-mediated autophagy involves the upregulation of Beclin1, an essential member of the PIK3C3 autophagic nucleation complex [45]. These data indicate a direct negative regulation of autophagy by Hsp70 in NSCLC cell lines under basal biological conditions.

In the context of heat shock exposure, despite the hyperactivation of HSF1 and Hsp70, we observed robust autophagy activation. Mechanistically, similar to genetic or chemical overexpression of Hsp70, heat shock-induced HSF1 and Hsp70 were also associated with reduced levels of AMPK phosphorylation and ULK1, which is in line with previous reports showing the downregulation of AMPK under heat shock treatment [43]. However, the heat shock-induced stress was associated with elevated levels of Beclin1, suggesting that autophagy induced upon heat shock exposure is a non-canonical autophagy that is independent of the AMPK-ULK1 pathway but dependent on Beclin1. In this regard, Beclin1 may be upregulated by a different mechanism, which could potentially involve the unfolded protein response or another Beclin1-activating kinase [45,46]. For instance, under glutamine deprivation, phosphoglycerate kinase 1 (PGK1) was shown to phosphorylate Beclin1 to induce autophagy, independent of AMPK- and ULK1-mediated phosphorylation of Beclin1 [47]. Similarly, autophagy activation upon ammonia treatment was shown to be independent of AMPK, TSC2, and ULK1, but dependent on both VPS34 acetylation and Beclin1 activation [48].

On the other hand, autophagy induction by rapamycin was associated with reduced levels of basal and heat shock-induced Hsp70. However, this effect may be mediated by mTORC1 suppression by rapamycin rather than autophagy induction itself, as mTORC1 was demonstrated to enhance HSF1 activation by direct phosphorylation on Ser326 [42]. In addition, AMPK upregulation following rapamycin treatment may also contribute to Hsp70 downregulation by suppressing HSF1-dependent Hsp70 upregulation [43]. To understand the effect of autophagy inhibition on Hsp70 expression levels or function, we used SAR405, a small-molecule compound and potent autophagy inhibitor that suppresses the catalytic activity of VPS34 kinase [25]. Our results demonstrate no significant effects of SAR405-mediated suppression on autophagy Hsp70 expression levels or function, which may suggest the recruitment of another compensatory degradation mechanism upon autophagy inhibition by SAR405, such as the ubiquitin–proteasome system.

A growing body of evidence has demonstrated that autophagy exhibits controversial effects on tumor cell survival or death, which are closely linked with the interplay of autophagy and apoptosis [49]. Therefore, we attempted to translate the impact of Hsp70 and autophagy mutual regulation on NSCLC cells physiology and response to chemotherapy to determine whether they adapt to Hsp70 inhibition by engaging autophagy.

Accordingly, we found that combining cisplatin and Hsp70 inhibitors (VER-155008, or JG-98) with early-stage (SAR405) or late-stage (CQ) autophagic inhibitors resulted in a robust reduction in the metabolic, clonogenic, and proliferative activities of tumor cells, as well as in cell viability in 2D and 3D spheroid models, which suggests an induction of pro-survival autophagy following Hsp70 inhibition. Furthermore, these tumor-suppressing effects upon autophagy inhibition were synergetic and exerted through potentiating apoptotic cell death. In support of these findings, it was shown that Hsp70 inhibition by VER-155008 enhanced autophagy activation in mesothelioma tumor cells; however, combining cisplatin with only VER-155008 failed to induce a synergetic effect on tumor cell death [50]. Previous reports have also demonstrated that tumor cells resistant to Hsp70 inhibition had an increased activation of autophagy as a compensatory strategy, whereby the use of CQ diminished tumor cell resistance to the Hsp70 inhibitor MAL3-101 [51,52]. In addition, the study of Wang et al. showed that A549 cells resistant to cisplatin were characterized by enhanced levels of autophagy and other proteostasis mechanisms such as Hsp70, UPR, and anti-apoptotic proteins [37].

Consistently, we also demonstrated that activating autophagy using rapamycin instead of autophagy inhibition, although it resulted in reduced metabolic and proliferative activity in NSCLC cells, indeed increased tumor cell viability and decreased apoptosis, as evaluated through Annexin/PI staining. This could be explained by the fact that rapamycin-mediated mTOR inhibition and autophagy activation, while reducing the proliferative and metabolic activity of tumor cells, may promote the enrichment of dormant cancer cells that remain viable but exhibit low proliferative and metabolic activity [15,53]. It is noteworthy that excessive autophagy activation may also induce apoptosis [54], reinforcing that modulating autophagy is a double-edged sword that is strictly dependent on the context, tumor type, and timing. These observations indicate that exploring the role of autophagy in each specific context is crucial to enhance our understanding and accurately predict the outcome of its modulation.

## 4. Materials and Methods

### 4.1. Cell Culture

Human lung adenocarcinoma A549 and H1299 cell lines were obtained from the shared research facility “Vertebrate cell culture collection” supported by the Ministry of Science and Higher Education of the Russian Federation (Agreement No. 075-15-2021-683). Tumor cells were cultivated in DMEM media supplemented with 10% heat inactivated fetal bovine serum (FBS) (HyClone, Logan, UT, USA), 2 mM L-glutamine, 100 U/mL penicillin, and 0.1 mg/mL streptomycin (PanEco, Moscow, Russia) at 37 °C in a 5% CO_2_ atmosphere with 90% humidity.

### 4.2. Plasmids

The transfer plasmid pGFP-C-shLenti, used for the knockdown of Hsp70 and its master regulator HSF1, was purchased from OriGene Technologies (Rockville, MD, USA). The specific clones used were TRCN0000008513 (shRNA against HSPA1A (Hsp70)), containing the mature sense sequence TTGATGCTCTTTGTTCAGGTCG, and TRCN0000280463 or TRCN0000007481 (shRNA against HSF1), containing the mature sense sequences CAAACGTGGAAGCTGTTCC and ATACTTGGGCATGGAATGTGC, respectively. The packaging plasmid pMD2.G and the viral envelope plasmid psPAX were purchased from Addgene (Watertown, MA, USA).

The plasmid expressing the Hsp70-cDNA gene was purchased from Genechem Co. (Shanghai, China). H1299 cells were seeded into 6-well culture plates to reach 60–70% at the day of transfection, then transfected with the plasmid using Lipofectamine^®^ 3000 Reagent (Thermo Fisher Scientific, Inc., Waltham, MA, USA), according to the manufacturer’s instructions.

### 4.3. Chemicals and Drug Treatments

VER-155008, SAR405, rapamycin, and chloroquine were obtained from Selleckchem, Houston, TX, USA. Cisplatin and JG-98 were purchased from Sigma-Aldrich, St. Louis, MO, USA. For MTT, western blotting, PCR, and Annexin V apoptosis assay, cells were seeded at the appropriate density to reach 70–80% confluency the next day (10 × 10^3^ cells for MTT in a 96-well plate, 300 × 10^3^ cells in a 6-well plate for Annexin V apoptosis assay, and 800 × 10^3^ cells in 5 cm cultivation dishes for western blotting and RT-PCR). The next day, cells were treated with the required concentrations for each experiment and then incubated for 24 h or 48 h, depending on the experiment conditions.

### 4.4. Heat Shock Treatment

To expose tumor cells to heat shock, 800 × 10^3^ tumor cells were seeded on 5 cm dishes for 24 h, then the dishes were securely sealed with parafilm and submerged directly in a water bath that was previously heated to 43 °C and left for 1 h. Then, cells were re-incubated again at 37 °C in a 5% CO_2_ incubator to recover, either for 5 or 18 h. Rapamycin treatment (100 nM) was started 2 h prior to heat shock exposure when needed.

### 4.5. Western Blot Analysis

Following the required manipulation of each experiment, tumor cells at a confluency of 70–80% were lysed using RIPA buffer (50 mM Tris-Cl pH 7.4, 1% NP-40, 150 mM NaCl, and 1 mM EDTA (purchased from Sigma-Aldrich, St. Louis, MO, USA)) supplemented with protease and phosphatase inhibitors (1 mM PMSF, 1 mM EGTA, 10 mM NaF, and 1 mM Na3VO4 (purchased from Sigma-Aldrich, St. Louis, MO, USA)), homogenized, and incubated for 15 min on ice. Protein extracts were obtained by centrifuging at 13,000 rpm, 4 °C, for 15 min. The concentration of protein samples was quantified using the Pierce™ BCA Protein Assay Kit (Thermo Fisher Scientific, USA). Protein samples (30 μg) were separated via SDS/polyacrylamide gel electrophoresis and transferred to a nitrocellulose membrane. After blocking non-specific antibody binding sites with 5% dry fat-free milk, the membrane was incubated overnight at 4 °C with primary antibodies against Hsp70 (clone 2B11), ATG5 (Cell signaling, Danvers, MA, USA #2630), LC3 I/II (Cell signaling #4108), p62 (Abcam, Cambridge, UK, #ab155686 or Servicebio, Wuhan, China, #GB11239-1-100), ULK1 total (Cell signaling #8054), p-ULK1 Ser555 (Cell signaling #5869), p-ULK1 Ser757 (Cell signaling #6888), p-AMPK (Abcam # ab133448), p-mTOR (Cell signaling #2971), β-Actin (Invitrogen, Waltham, MA, USA, #MA1-744), and β-tubulin (Invitrogen, #32-2600). After that, membranes were washed three times with PBS-T and incubated with peroxidase-conjugated secondary antibodies for 1 h. They were then washed four times with PBS-T, and then the chemiluminescence of protein bands was detected using the ChemiDoc imaging system (Bio-Rad, Hercules, CA, USA).

### 4.6. Hsp70 Substrate-Binding Assay

The assessment of the substrate-binding ability of Hsp70 was performed as described previously in [20]. Briefly, cells were seeded in 6-well plates, treated with different compounds at the required concentrations, and incubated for 24 h before lysing, collecting, and measuring the protein concentration of the cell lysates. Next, 10 μg/mL of carboxymethylated lactalbumin (CMLA) in PBS was added to the wells of a 96-well plate, followed by the blocking of non-specific bindings using 5 mg/mL bovine serum albumin in PBS. After that, cell lysates were added to the wells with CMLA and incubated for 2 h. Next, wells were washed, and rabbit polyclonal antibodies to Hsp70 were added, followed by secondary anti-rabbit antibodies conjugated with peroxidase (Jackson Immunochemicals, Tucker, GA, USA). Finally, tetramethylbenzene in a citrate buffer (pH 4.5) with hydrogen peroxide was added, and the staining degree was measured using a Varioskan LUX Multimode Microplate Reader (Thermo Fisher Scientific, USA).

### 4.7. Cell Viability Assay (MTT Assay)

Cells were seeded at a density of 10,000 cells/well in a 96-well plate, and following 24 h, they were treated with the indicated drugs in accordance with the indicated experimental conditions. To quantify cell viability, 100 µL of MTT (PanEco, Russia) working solution was added to the cells and incubated for 2 h. Then, MTT solution was aspirated, the formed formazan crystals were dissolved in DMSO (BioloT, St. Petersburg, Russia), and the absorbance was measured at 570 nm using a Varioskan LUX Multimode Microplate Reader (Thermo Fisher Scientific, USA).

### 4.8. Analysis of Drugs Combination Index

Based on the MTT assay data, the drug combination index (CI) was calculated using the formula: CI = ABC/(A + B + C). In this formula, ABC denotes the ratio of cells killed under the combination of three drugs compared to the control group (untreated), while A, B, and C represent the ratios of the killed cells under each single agent compared to the control group. A CI value of less than 1 suggests drug synergy; a value equal to 1 suggests an additive effect; and a value greater than 1 indicates drug antagonism.

### 4.9. Real-Time PCR

Total RNA was extracted from cells using ExtractRNA Reagent (Evrogen, Moscow, Russia) according to the manufacturer’s protocol, and 1 μg of RNA was reverse-transcribed using the MMLV RT kit (Evrorgen, Russia). Then, gene expression was analyzed using the qPCRmix-HS SYBR kit (Evrogen, Russia) according to the manufacturer’s recommendations. All RT-PCR studies were performed using the CFX96 Real-Time detection system (Bio-Rad, USA). Relative gene expression to Actin was then calculated using the ΔΔCt method. The sequences of primers used for qPCR were as shown in Table 1.

### 4.10. Colony Formation Assay

A549 or H1299 cells were seeded in the wells of a 6-well plate at a concentration of 1000 cells per well. After 24 h, the cells were treated with 2 µM of VER-155008, SAR405, and cisplatin, or 20 nM Rapamycin, as well as their indicated combinations for 48 h. Then, cells were gently washed twice with PBS, fresh full medium was added, and cells were incubated for 8–10 days in 5% CO_2_ at 37 °C. After that, the grown colonies were fixed with 10% formaldehyde and stained with 0.2% crystal violet. The plate was dried and scanned using the ChemiDoc imaging system (Bio-Rad, USA).

### 4.11. Annexin V/PI Staining Apoptotic Assay

NSCLC cells were seeded at a concentration of 300 × 10^3^ cells/well for 24 h and then treated with cisplatin (15 µM) alone and in the indicated combinations with VER-155008 (15 µM), SAR405 (15 µM), and rapamycin (150 nM), followed by a 48 h incubation. Next, cells were harvested, washed with cold PBS, resuspended in 250 μL of Annexin V-binding buffer, and then stained with Annexin V Alexa 647 and Propidium Iodide (Thermo Fisher Scientific, USA), according to the manufacturer’s recommendations, for 15 min. Finally, cells were diluted to 500 μL of Annexin V-binding buffer and immediately analyzed using CytoFlex FACS Flow cytometer (Beckman Coulter, Brea, CA, USA). The results were analyzed using CytExpert 2.0 software (Beckman Coulter, Brea, CA, USA).

### 4.12. CellTiter-Glo 3D-Spheroid Viability Assay

NSCLC cell-seeding, the formation of spheroids, and the 3D CellTiter-Glo viability assay were performed as described previously [55]. Briefly, a 1% agarose solution diluted in DMEM was microwaved to complete the dilution, then 30 μL was added to the wells of 96-well plates and left to cool down and solidify. Next, NSCLC cells were seeded at a concentration of 12 × 10^3^ cells/well, and the plate was centrifuged for 45 min at 3000 rpm and then incubated for 72 h at 37 °C in a 5% CO_2_ incubator to form 3D spheroids. The next day, chemical compounds were added to the wells at the indicated concentration, and all spheroids were captured using a microscope camera every two days at a 40× magnification. On the sixth day, the 3D spheroids were either carefully pipetted (2–3 times) to assess the degree of cell aggregation and spheroid integrity, or tumor cell viability was assessed using the 3D CellTiter-Glo^®^ Luminescent cell viability assay (Promega, Madison, WI, USA), according to the manufacturer’s protocol.

### 4.13. Statistical Analysis

Numerical results are reported as the mean ± standard error of the mean (SEM) and represent data from three or four replicates of three independent experiments. Quantitative analysis was performed with the use of Graph Pad Prism 9.5 (Graph Pad Software Inc., San Diego, CA, USA). One-way or two-way ANOVA followed by Tukey’s or Dunnett’s multiple comparison tests were used depending on the type of experiment. Differences were considered statistically significant at *p* < 0.05.

## 5. Conclusions

Our study provides new insights into the interplay between Hsp70 and autophagy in NSCLC cells and highlights the employment of proteostasis mechanisms crosstalk as a promising target in combinatorial therapies. In particular, we found that Hsp70 governs autophagy activation by negatively regulating AMPK activation, while AMPK-dependent autophagy is activated upon Hsp70 inhibition as a compensatory pro-survival strategy in NSCLC cells. Therefore, we propose that the simultaneous targeting of two interconnected cellular defense mechanisms, Hsp70 and autophagy, may be promising to overcome therapeutic resistance and diminish the subpopulations of dormant cancer cells responsible for the induction of metastasis and tumor recurrence [15]. The limitations of this study are that our findings were obtained *in vitro*; thus, these results should be recapitulated *in vivo*, and such a combinatorial therapy should be tested in animal models to verify its effectiveness. Another limitation is that the exact molecular mechanism by which Hsp70 influences the phosphorylation and activation of AMPK, mTOR, and ULK1 is not clear and needs to be determined through future investigations; this may involve the activity of upstream kinases like AKT [56]. Moreover, the development of more potent Hsp70 inhibitors is still required to reduce the high toxicity of currently available Hsp70 inhibitors [57,58].

Interestingly, the very recent study of Ferretti et al. has demonstrated similar findings in pancreatic cancer, whereby Hsp70 inhibition resulted in the activation of AMPK–Beclin1-mediated autophagy, while the combined inhibition of Hsp70 and autophagy exerted synergetic effects on pancreatic tumor cell viability, both in vitro and in vivo [59]. Altogether, the combination therapies targeting both Hsp70 and autophagy may offer a promising strategy to diminish therapeutic resistance in several refractory cancers and, furthermore, may enable a decrease in the dose of first-line chemotherapeutic drugs, potentially reducing therapeutic cytotoxicity and achieving better clinical responses in lung and other cancers.

## Figures and Tables

**Figure 1 ijms-25-09090-f001:**
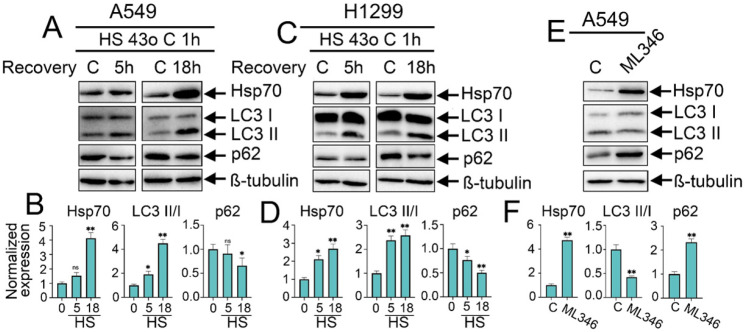
Effect of heat shock exposure and Hsp70 chemical overexpression on autophagy activation in NSCLC tumor cells. (**A**,**C**) Human A549 and H1299 tumor cells were exposed to heat shock for 1 h at 43 °C, and incubated for 5 or 18 h to recover. Then, cells were lysed and subjected to immunoblotting with antibodies against Hsp70, p62, and LC3 I/II. (**E**) A549 tumor cells were treated with ML346 at a concentration of 10 µM for 24 h and then were lysed for immunoblotting. (**B**,**D**,**F**) Values on the charts represent the relative protein expression in the respective sample, which indicates the ratio between the band intensity of the protein of interest and the band intensity of β-Tubulin. Band intensity was measured using the ImageJ software 1.53. The results were considered statistically significant at * *p* < 0.05, ** *p* < 0.001. ns: non-significant, HS: Heat shock.

**Figure 2 ijms-25-09090-f002:**
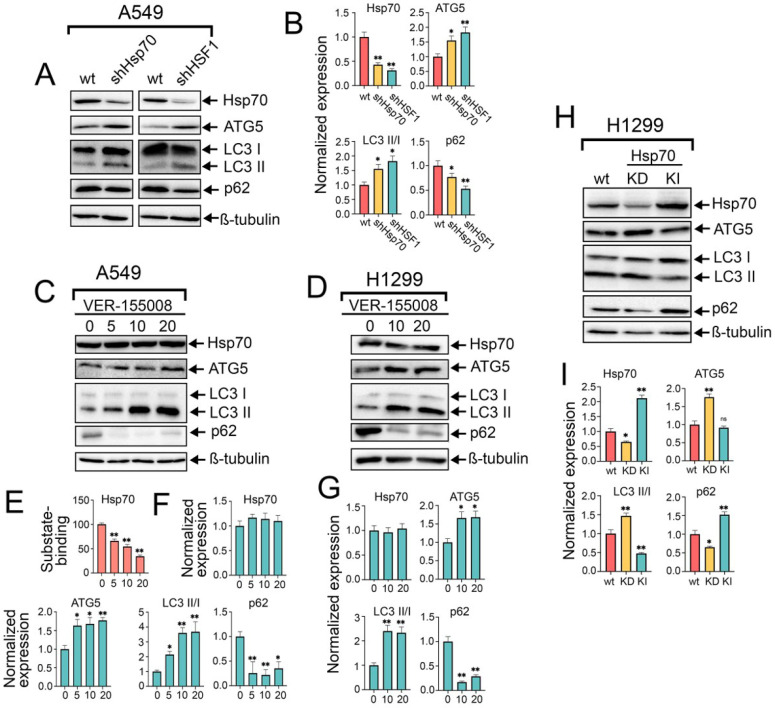
The knockdown of HSF1 or Hsp70, as well as the use of VER-155008, activates autophagy, while Hsp70 knock-in inhibits autophagy in NSCLC tumor cells. (**A**) A549 control cells (sh-scrambled) and A549 (shHsp70 or shHSF1) were cultivated until 70–80% confluency, then lysed and subjected to immunoblotting with antibodies against HSP70, ATG5, p62, and LC3 I/II. (**C**,**D**) NSCLC cell lines were treated with VER-155008 at the indicated concentrations for 24 h. After that, cells were lysed and subjected to immunoblotting with antibodies against Hsp70, ATG5, p62, and LC3 I/II. (**E**) A549 tumor cells were treated with VER-155008 at the indicated concentration and then subjected to Hsp70 substrate-binding assay. (**H**) H1299 tumor cells with shHsp70 or Hsp70 knock-in were cultivated until 70–80% confluency and then subjected to western blotting using antibodies against Hsp70, ATG5, p62, and LC3 I/II. (**B**,**F**,**G**,**I**) Values on the charts represent the relative protein expression in the respective sample, which indicates the ratio between the band intensity of the protein of interest and the band intensity of β-Tubulin. Band intensity was measured using the ImageJ software 1.53. Values are the means ± SEM from three independent experiments. * *p* < 0.05, ** *p* < 0.001. ns: non-significant.

**Figure 3 ijms-25-09090-f003:**
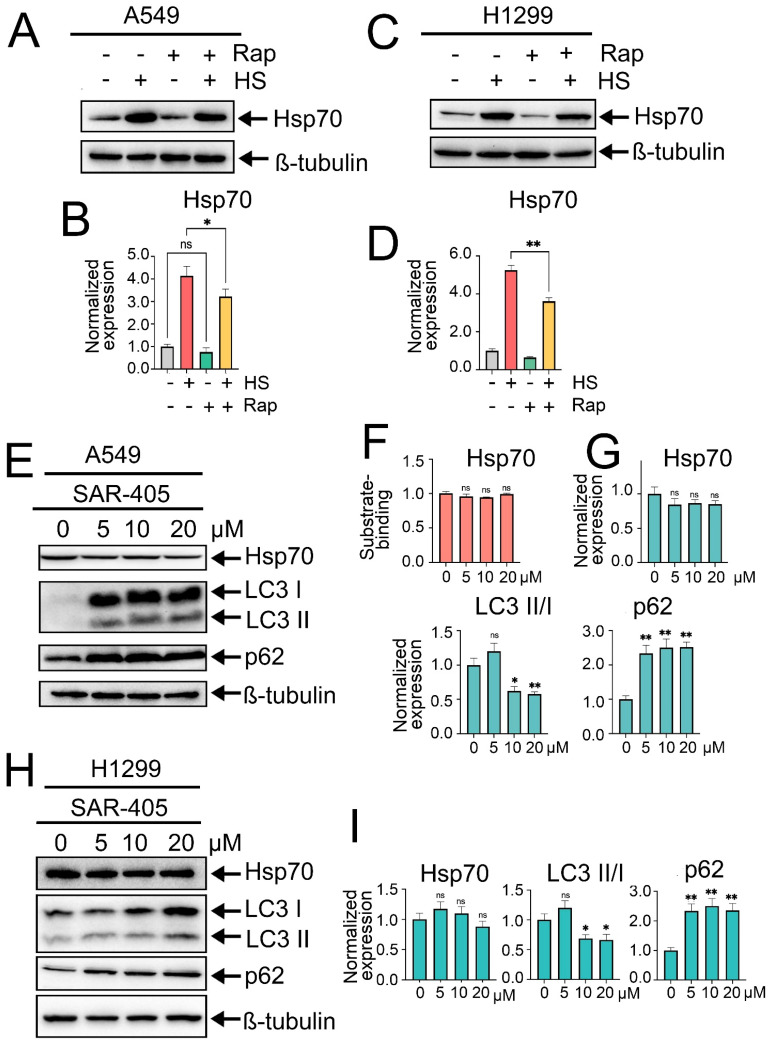
Effects of rapamycin-induced autophagy or autophagy inhibition by SAR405 on Hsp70 activation. (**A**,**C**) A549 and H1299 tumor cells were either pretreated or not with 100 nm rapamycin for 2 h and then exposed to heat shock for 1 h at 43 °C, followed by incubation for 18 h to recover. After that, cells were lysed and subjected to immunoblotting with antibodies against Hsp70. (**E**,**H**) A549 and H1299 tumor cell lines were treated with SAR405 at specified concentrations for 24 h. Following treatment, the cells were lysed and analyzed by immunoblotting using antibodies against Hsp70, ATG5, p62, and LC3 I/II. (**F**) A549 tumor cells were treated with SAR405 at the specified concentration, then they were lysed and subjected to Hsp70 substrate-binding assay. (**B**,**D**,**G**,**I**) Values on the charts represent the relative protein expression in the respective sample, which indicates the ratio between the band intensity of the protein of interest and the band intensity of β-Tubulin. Band intensity was measured using the ImageJ software 1.53. Values are the means ± SEM from three independent experiments; * *p* < 0.05, ** *p* < 0.001. ns: non-significant.

**Figure 4 ijms-25-09090-f004:**
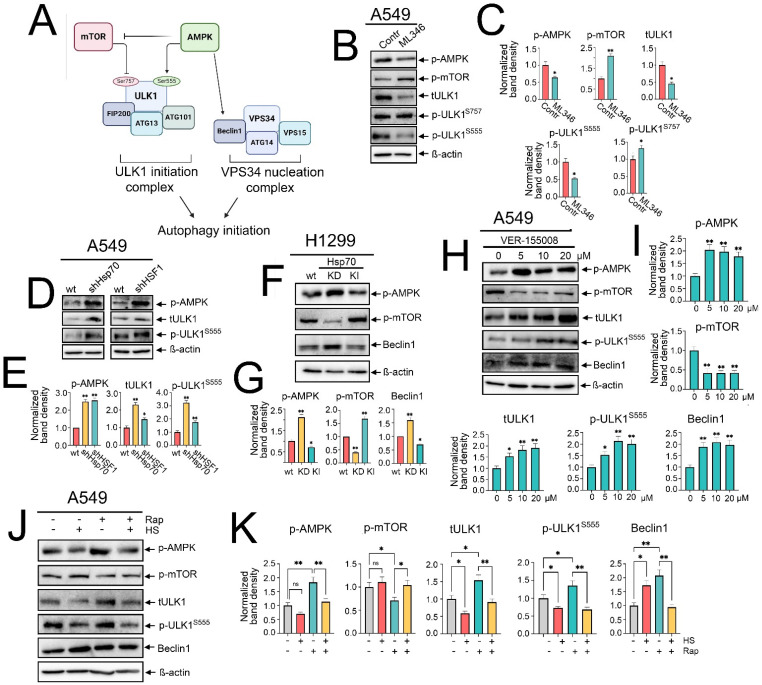
Hsp70 regulates autophagy activation by influencing AMPK/mTOR interplay and ULK1/Beclin1 activation. (**A**) A representative scheme of the regulation of autophagy imitation by AMPK/mTOR activation and the involvement of ULK1 and VPS34 complexes. (**B**) A549 tumor cells were treated with ML346 at a concentration of 10 µM for 24 h, and then were lysed for immunoblotting with antibodies against p-AMPK, p-mTOR, total ULK1, p-ULK1 Ser555, and p-ULK1 Ser757. (**D**) A549 control cells (sh-scrambled) and A549 (shHsp70 or shHSF1) or (**F**) H1299 tumor cells with shHsp70 or Hsp70 knock-in were cultivated until 70–80% confluency, then lysed, and immunoblotting was performed. (**H**) A549 tumor cells were treated with VER-155008 at the indicated concentrations and then were lysed for immunoblotting. (**J**) A549 tumor cells were pretreated or not with 100 nM rapamycin for 2 h, then subjected to heat shock at 43 °C for 1 h, followed by an 18 h recovery incubation. Subsequently, the cells were lysed and analyzed by immunoblotting using antibodies against p-AMPK, p-mTOR, total ULK1, p-ULK1 Ser555, and Beclin1. (**C**,**E**,**G**,**I**,**K**) Values on the charts represent the relative protein expression in the respective sample, which indicates the ratio between the band intensity of the protein of interest and the band intensity of β-actin. Band intensity was measured using the ImageJ software 1.53. * *p* < 0.05, ** *p* < 0.001. ns: non-significant.

**Figure 5 ijms-25-09090-f005:**
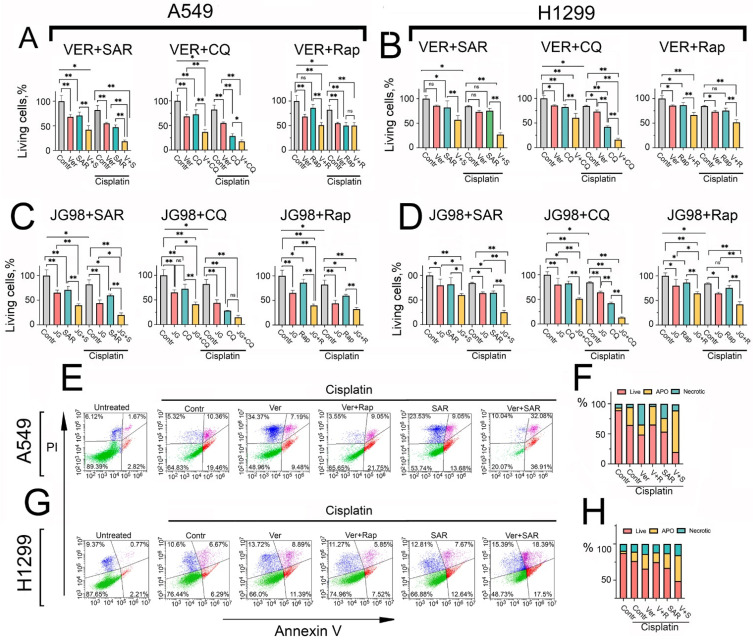
Hsp70 inhibition-induced autophagy plays a pro-survival role in NSCLC tumor cells, and combined inhibition of Hsp70 and autophagy with cisplatin synergistically induces apoptosis. (**A**,**B**) A549 and H1299 tumor cells were treated with combinations involving the following compounds: cisplatin (10 µM), VER-155008 (10 µM), JG-98 (1 µM), SAR405 (10 µM), CQ (40 µM), and rapamycin (0.1 µM) for 48 h. Afterwards, tumor cell viability was assessed using the MTT test. (**C**,**D**) A549 and H1299 tumor cells were treated with combinations of the following compounds: cisplatin (10 µM), JG-98 (1 µM), SAR405 (10 µM), CQ (40 µM), and rapamycin (0.1 µM) for 48 h. Following treatment, cell viability was assessed using the MTT assay. (**E**–**H**) A549 and H1299 tumor cells were treated with different combinations of the following compounds: cisplatin (15 µM), VER-155008 (15 µM), SAR405 (15 µM), and rapamycin (0.15 µM) for 48 h. Subsequently, cells were stained with Annexin V/PI for 30 min and directly analyzed using flow cytometry. Values are the means ± SEM from three independent experiments; * *p* < 0.05, ** *p* < 0.001. ns: non-significant.

**Figure 6 ijms-25-09090-f006:**
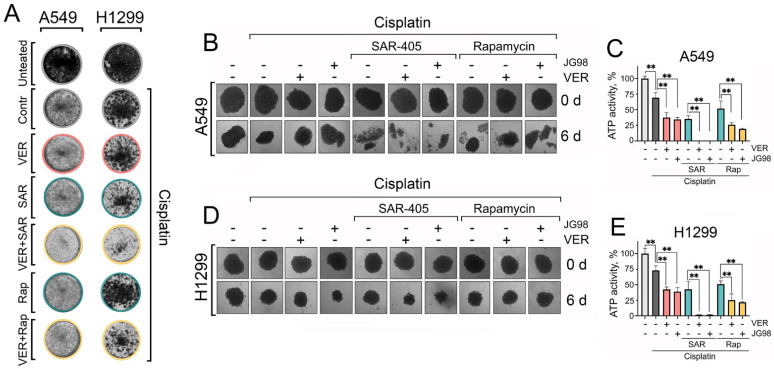
The dual inhibition of Hsp70 and autophagy with cisplatin NSCLC clonogenic potential and 3D tumor spheroids viability. (**A**) A549 and H1299 tumor cells were treated with combinations of cisplatin (2 µM), VER-155008 (2 µM), SAR405 (2 µM), and rapamycin (20 nM) for 48 h, then the treatments were removed and cells were washed and left to form colonies. After ten days for A549 cells or eight days for H1299 cells, they were fixed with 10% formalin and stained with 0.2% crystal violet. (**B**–**E**) A549 and H1299 were seeded on 1% agarose in 96-well plates for 3 days to form spheroids. Afterwards, they were treated with combinations of cisplatin (30 µM), VER-155008 (30 µM), SAR405 (30 µM), JG-98 (3 µM), and rapamycin (0.3 µM) for six days. The spheroids’ morphology was captured (microscope’s magnification ×40), and they were then subjected to a CellTiter-Glo 3D-Spheroid viability assay. Values are the means ± SEM from three independent experiments; * *p* < 0.05, ** *p* < 0.001.

**Table 1 ijms-25-09090-t001:** Primers used in this study.

Gene of Interest	Forward	Reverse
Hsp70	5′-AGAAGGACATCAGCCAGAACAA-3′	5′-AGAAGTCGATGCCCTCAAACA-3′
ULK1	5′-CTGCTGGGGAAGGAAATCAAAAT-3′	5′-AACCAGGTAGACAGAATTAGCCAT-3′
ATG7	5′-AAGCCATGATGTCGTCTTCCTAT-3′	5′-GCATTGATGACCAGCTTTCTCTT-3′
p62	5′-TCAGGAGGAGATGATGACTGGA-3′	5′-TTGGCCCTTCGGATTCTGG-3′
Actin	5′-CCATCATGAAGTGTGACGTGC-3′	5′-GTCCGCCTAGAAGCATTTGCG-3′

## Data Availability

All data are included within the article and its Appendix A.

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
