# Peer review of "Hsp70 Negatively Regulates Autophagy via Governing AMPK Activation, and Dual Hsp70-Autophagy Inhibition Induces Synergetic Cell Death in NSCLC Cells"

_ijms, 2024, doi:10.3390/ijms25169090_

Round 1
Reviewer 1 Report
Comments and Suggestions for Authors
The manuscript “Hsp70 negatively regulates autophagy via governing AMPK activation and dual targeting of Hsp70 and autophagy synergizes with cisplatin to induce apoptotic cell death in lung cancer” by Bashar Alhasan and co-authors to investigate the molecular interplay between Hsp70 and autophagy in lung cancer cells and elucidated its impact on the outcomes of anticancer therapies. We found that Hsp70 negatively regulates autophagy by directly influencing AMPK activation, uncovering a novel regulatory mechanism of autophagy by Hsp70. Genetic or chemical Hsp70 overexpression were correlated with inhibition of AMPK and autophagy, and conversely, AMPK-mediated autophagy is upregulated following both genetic and chemical Hsp70 inhibition. We further investigated whether Hsp70 suppression-mediated autophagy exhibits pro-survival or pro-death inducing effects. Our results show that combined inhibition of Hsp70 and autophagy, along with cisplatin treatment, synergistically reduces tumor cell metabolic activity, growth, and viability. These cytotoxic effects were exerted via substantially potentiating apoptosis, while activating autophagy via rapamycin slightly rescued tumor cells from apoptosis. Therefore, our findings demonstrate that the combined inhibition of Hsp70 and autophagy represents a novel therapeutic approach that may be promising in disrupting the capacity of refractory tumor cells to withstand conventional therapies in lung cancer. However, some concerns that must be taken into account before the work can be reconsidered for publication.
Comments
1. Author should explain why cell incubated for 5 or 18 hours to recover. Please cite reference.
2. Figure 6A: A whole plate of colonies formation should be added to supplementary.
3. Can author perform autophagy markers by fluorescence microscope to confirm these results?
Comments on the Quality of English LanguageModerate editing of English language required
Reviewer 2 Report
Comments and Suggestions for Authors
Your paper -Hsp70 negatively regulates autophagy via governing AMPK activation and dual targeting of Hsp70 and autophagy synergizes with cisplatin to induce apoptotic cell death in lung cancer, presents an interesting subject with increased importance in cancer field, but I suggest a few observations to improve the manuscript quality:
1.-Please, to increase the clarity and quality of your title, you must short their length, keeping the action about the key mechanisms, but also adding the cell cancer types (adenocarcinoma) and mention that are cancer cell cultures developed in vitro. The title highlights, in generally the manuscript importance, but yours is a long title. Please reduce its.
2.-Please, add your institutional e-mails of authors, and eliminate your ORCHID Ids. These are journal requirements.
3.- Please, revise entire abstract to highlights the background, materials and methods, obtained results, and conclusion.
4.- Please, write in italic mode “in vitro” and “in vivo”. Defines abbreviations in scientifically terms in your manuscript.
5. Please, add future directions of your research. Also, add limitations of this study.
6. Add, Conclusion section in your manuscript.
7.- In materials and methods section, please correct a few paragraph –lines 524-525; Correct the chemicals formula such as –CO2 and others. Please check its. Also, add the producer and country for all chemical substances.
Comments on the Quality of English LanguageMinor editing of English language required.
